# NAD+ prevents septic shock-induced death by non-canonical inflammasome blockade and IL-10 cytokine production in macrophages

**Jasper Iske[1,2†], Rachid El Fatimy[3,4†], Yeqi Nian[5†], Amina Ghouzlani[6], Siawosh K Eskandari[7], Hector Rodriguez Cetina Biefer[1,8], Anju Vasudevan[9], Abdallah Elkhal[1,6*]**

[1]Division of Transplant Surgery, Department of Surgery, Brigham and Women's Hospital, Harvard Medical School, Boston, United States; [2]Department of Cardiothoracic and Vascular Surgery, Germany Heart Center Berlin, Berlin, Germany; [3]Department of Neurology, Ann Romney Center for Neurologic Diseases, Brigham and Women's Hospital, Harvard Medical School, Boston, United States; [4]Institute of Biological Sciences (ISSB-P), Mohammed VI Polytechnic University, Benguerir, Morocco; [5]Institute of Transplant Medicine, Tianjin First Central Hospital, Nankai University, Tianjin, China; [6]NAD[6] Immunology Laboratory, Huntington Medical Research Institutes, Pasadena, United States; [7]Department of Internal Medicine, University of Groningen, Groningen, Netherlands; [8]Department of Cardiac Surgery, Stadtspital Zurich Triemli, Zurich, Switzerland; [9]Department of Neurosciences, Angiogenesis and Brain Development Laboratory, Huntington Medical Research Institutes, Pasadena, United States

**\*For correspondence:**
Abdala.Elkhal@hmri.org

†These authors contributed equally to this work

**Competing interest:** The authors declare that no competing interests exist.

**Abstract** Septic shock is characterized by an excessive inflammatory response depicted in a cytokine storm that results from invasive bacterial, fungi, protozoa, and viral infections. Non-canonical inflammasome activation is crucial in the development of septic shock promoting pyroptosis and proinflammatory cytokine production via caspase-11 and gasdermin D (GSDMD). Here, we show that NAD+ treatment protected mice toward bacterial and lipopolysaccharide (LPS)-induced endotoxic shock by blocking the non-canonical inflammasome specifically. NAD+ administration impeded systemic IL-1β and IL-18 production and GSDMD-mediated pyroptosis of macrophages via the IFN-β/STAT-1 signaling machinery. More importantly, NAD+ administration not only improved casp-11 KO (knockout) survival but rendered wild type (WT) mice completely resistant to septic shock via the IL-10 signaling pathway that was independent from the non-canonical inflammasome. Here, we delineated a two-sided effect of NAD+ blocking septic shock through a specific inhibition of the non-canonical inflammasome and promoting immune homeostasis via IL-10, underscoring its unique therapeutic potential.

## eLife assessment

In this **valuable** contribution, the authors demonstrate that the infusion of NAD+ may prevent death and reduce disease severity from lethal experimental bacterial sepsis, possibly through inflammasome inhibition, without reducing bacterial load. They provide **solid** evidence for these protective effects of NAD+, though the precise mechanisms involved remain unclear and need further support and elucidation. The core findings may well have clinical implications but, in addition to mechanistic

clarifications, contextualised interpretation as metabolic adaptation to sepsis would create wider interest.

## Introduction

Sepsis is characterized by a systemic inflammatory response syndrome (*Kaukonen et al., 2015*) driven by host cells following systemic bacterial (*Ramachandran, 2014*) and viral infections. The excessive inflammatory response can derail into septic shock resulting in multiple organ failure, the leading cause of death in intensive care units. Inflammasome activation, which downstream pathways cause the release of proinflammatory cytokines and the induction of an inflammatory cell death termed pyroptosis (*Kumar, 2018*), has been pointed out as the major driver of septic shock. Hereby, a two-armed lipopolysaccharide (LPS)-derived induction of the NLRP3-canonical inflammasome, the major source of IL-1β and IL-18 cytokine production (*Lopez-Castejon and Brough, 2011*) and the caspase-11-mediated non-canonical inflammasome leading to pyroptosis in monocytes (*Yi, 2017*), was determined as the underlying mechanism. Mechanistically, caspase-11 acts as a pattern recognition receptor for intracellular bacteria (*Ding and Shao, 2017*) that cleaves gasdermin D (GSDMD), a membrane pore-forming protein subsequently inducing pyroptotic cell death (*Shi et al., 2015*). The NLRP3-canonical inflammasome in turn was found to be indispensable (*Man et al., 2017*) for septic shock-induced death. However, cross-activation through caspase-11 promoting cytokine release has been described (*Kayagaki et al., 2015*; *Kayagaki et al., 2011*; *Yang et al., 2015a*), assigning the non-canonical inflammasome a cardinal role (*Kayagaki et al., 2013*).

Recent approaches such as anti-proinflammatory cytokine strategies blocking downstream targets of inflammasomes have been ineffective (*Angus and van der Poll, 2013*) while inhibiting inflammatory key regulators such as NF-κB may promote adverse side-effects (*Fraser, 2006*). Hence, contemporary clinical therapy of septic shock is based on symptomatic treatment rather than curative approaches that clear the cause of the disease itself.

In our previous studies, we have underscored the immunosuppressive properties of NAD+ in autoimmune diseases and allo-immunity via the regulation of CD4+ T cell fate (*Tullius et al., 2014*; *Elkhal et al., 2016*). More recently, we have shown that NAD+ administration protected mice from lethal doses of *Listeria monocytogenes* (L. m.) via mast cells (MCs) exclusively and independently of major antigen presenting cells (APCs) (*Rodriguez Cetina Biefer et al., 2018*). However, the underlying mechanism that allows NAD+, to concomitantly protect against autoimmune diseases, via its immunosuppressive properties (*Tullius et al., 2014*; *Elkhal et al., 2016*), and against lethal bacterial infection remains unclear.

Therefore, in the current study we investigated whether NAD+ protects against bacterial infection by dampening the systemic inflammatory response associated with sepsis or through enhanced bacterial clearance. Although, wild type (WT) mice subjected to NAD+ or PBS and lethal doses of pathogenic *Escherichia coli* (*E. coli*) exhibited similar bacterial load in various tissues, mice treated with NAD+ displayed a robust survival. Moreover, NAD+ protected against LPS-induced death that was associated with a dramatic decrease of systemic IL-1β and IL-18 levels, two major cytokines involved in the inflammasome signaling machinery. More importantly, we show that NAD+ protected from LPS-induced death by targeting specifically the non-canonical inflammasome via a blockade of the STAT1/IFN-β signaling pathway. Moreover, NAD+ treatment rendered not only caspase-11 knockout (KO) but WT mice fully resistant to poly(I:C)+LPS-induced septic shock, via an inflammasome-independent pathway mediated by a systemic IL-10 cytokine production.

## Results

### NAD+ protects mice against septic shock not via bacterial clearance but via inflammasome blockade

Our previous studies have underscored the role of NAD+ in regulating CD4+ T cell fate and its immunosuppressive properties via IL-10 cytokine production (*Tullius et al., 2014*; *Elkhal et al., 2016*; *Rodriguez Cetina Biefer et al., 2018*). More recently, we have shown that NAD+ protected mice against lethal doses of L. m. independently of major APCs (*Tullius et al., 2014*). However, it remained

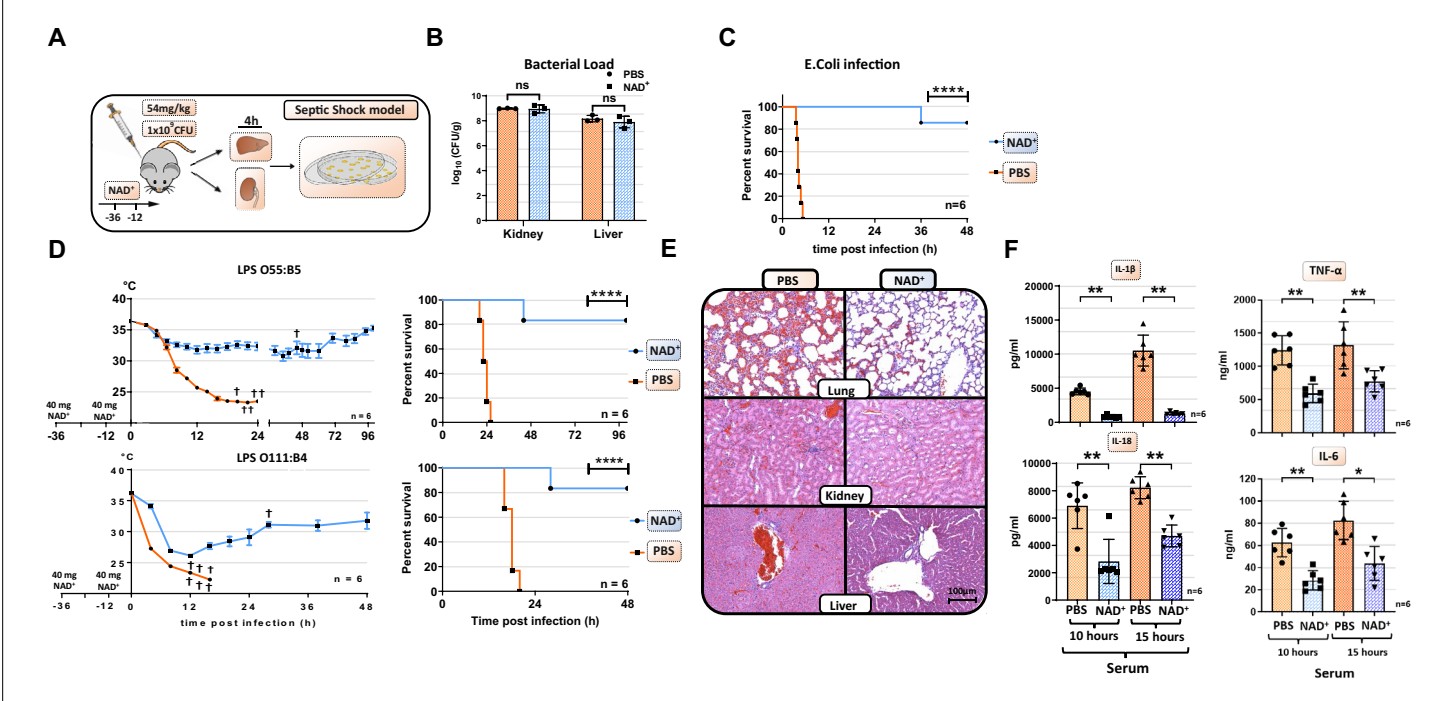

**Figure 1.** NAD+ protects mice from lethal bacterial infection and endotoxic shock by dampening systemic inflammation. (**A**) Mice were treated with PBS or NAD+ prior to administration of a lethal dose of either pathogenic *E. coli* or lipopolysaccharide (LPS) by intraperitoneal injection. (**B**) After the death of each animal, lungs, kidney and livers were removed and bacterial load was determined by counting colony-forming unit (CFU). Column plots display mean with standard deviation (n=3). (**C**) Survival was monitored over 48 hr after bacterial infection and (**D**) LPS injection of both serotypes (n=6, 3 independent survival experiments). In addition, body temperature was monitored in the kinetics of up to 100 hr. (**E**) Lungs, kidneys, and livers were removed and IHC was performed for hematoxylin and eosin (H&E) staining. (**F**) Systemic levels (serum) of IL-6, TNFα, IL-1β, and IL-18 were assessed by ELISA. Column plots display mean with standard deviation (n=5). Statistical significance was determined by using Student's t-test or one-way ANOVA while survival data were compared using log-rank Mantel-Cox test. Asterisks indicate p-values *=p<0.05, **=p<0.01, and ***=p<0.001, only significant values are shown. All data depicted in this figure are provided as source data.

The online version of this article includes the following source data and figure supplement(s) for figure 1:

**Source data 1.** Raw data for *Figure 1B*: Bacterial load.

**Source data 2.** Raw data for *Figure 1C*: *E. coli* infection.

**Source data 3.** Raw data for *Figure 1D*: Lipopolysaccharide (LPS) infection.

**Source data 4.** Raw data for *Figure 1E*: Histology.

**Source data 5.** Raw data for *Figure 1F*: ELISA.

**Figure supplement 1.** NAD+ preserves ileal villi structure and reduces splenic hemorrhage during lipopolysaccharide (LPS)-induced septic shock.

**Figure supplement 1—source data 1.** Raw data for *Figure 1—figure supplement 1*: Histology.

**Figure supplement 2.** Neutrophils per mm² infiltrating mice: ileum, kidney, lung, and liver in the IHC stains.

**Figure supplement 2—source data 1.** Raw data for *Figure 1—figure supplement 2*: Neutrophil count.

unclear whether NAD+ protected mice against lethal doses of L. m., a gram-positive bacterium, via a clearance mechanism or by dampening the inflammatory response. Since L. m. is known to be an intra-cellular pathogen, we tested if NAD+ protects as well against *E. coli*, a gram-negative bacterium that is well known to induce septic shock (*Mellata et al., 2016*). WT mice were treated with NAD+ or PBS for 2 consecutive days followed by a lethal dose (1×10⁹) of *E. coli* or PBS. Notably, mice treated with PBS died within 5 hr after infection, while mice treated with NAD+ exhibited an impressive survival (*Figure 1A*). Moreover, when assessing the bacterial load in liver and kidney (*Figure 1B*), organs exposed to the infection, by counting CFU in both, NAD+ and PBS groups, revealed no significant difference, suggesting that NAD+ does not promote bacterial clearance. More importantly, these data suggest that NAD+ may reduce the inflammatory response toward bacterial infection. It is well established that the bacterial LPS abundant on the outer membrane exhibits a key role in the pathology

of *E. coli*-derived septic shock (*Angus and van der Poll, 2013*). Thus, we further characterize the impact of NAD$^+$ on septic shock by subjecting mice to a lethal dose (54 mg/kg) of two different LPS serotypes (O111:B4 and O55:B5) described to vary in the antigen lipid A content and to promote distinct hypothermia kinetics (*Dogan et al., 2000*). Following LPS (O111:B4 and O55:B5) administration, PBS-treated control mice displayed severe symptoms of endotoxic shock with a dramatical body temperature decrease (<23°C) within 15 hr. In contrast, mice subjected to NAD$^+$ exhibited highly distinct kinetics with a recovery of body temperatures after 15 hr (*Figure 1C*). When monitoring survival, 100% of PBS-treated mice succumbed to LPS after 24 hr while NAD$^+$-treated animals exhibited an overall survival >85% (*Figure 1D*), which was consistent with our bacterial infection model. Mice infected and treated with NAD$^+$ survived for several months and recovered fully after 10 days. Of note, mice survived for over a year following infection and died of aging. LPS-induced death derives from multi-organ failure (*Bullock and Benham, 2019*). Therefore, lung, kidney, liver, ileum, and spleen were harvested 15 hr after LPS administration and tissue damage was assessed by hematoxylin and eosin (H&E) staining. Tissue evaluation indicated severe pulmonary hemorrhage, excessive tubular fibrin deposition, hepatocyte cell swelling, disseminated intravascular coagulation (DIC), and ileal villi destruction consistent with a multi-organ dysfunction syndrome (*Rossaint and Zarbock, 2015*) in mice treated with PBS. In contrast, NAD$^+$ administration dramatically attenuated signs of organ failure with significantly less pulmonary hemorrhage and DIC, intact liver and kidney tissue architecture, and preserved ileal villi (*Figure 1E*, *Figure 1—figure supplement 1*, and *Figure 1—figure supplement 2*). To elucidate the protective effects of NAD$^+$ systemic levels of IL-1β and IL-18, two major cytokines implicated in inflammasome activation were measured 10 and 15 hr after intraperitoneal injection of LPS (*Figure 1F*). Of, note IL-6 and TNFα systemic levels were measured as well (*Figure 1F*). Our findings indicated that LPS injection resulted in a robust systemic increase of IL-1β, IL-6, TNFα, and IL-18 in the PBS group, which was almost abolished in NAD$^+$-treated mice. Taken together, our results suggest that NAD$^+$ protects mice against septic shock not via bacterial clearance but rather via inflammasome blockade.

## NAD$^+$ specifically inhibits the non-canonical inflammasome

Our data suggest that NAD$^+$ protects against septic shock via inflammasome blockade. Monocytes, especially macrophages, have been described as major drivers of inflammasome-derived cytokine secretion in the context of septic shock (*Evans, 1996*). Thus, to test the effect of NAD$^+$ on inflammasome function, bone marrow-derived macrophages (BMDMs) were obtained and both canonical and non-canonical inflammasomes were stimulated in the presence or absence of 100 μmol/ml NAD$^+$. Activation of the canonical pathway was achieved through LPS priming (1 μg/ml) followed by ATP stimulation (5 mmol/l). Notably, BMDMs subjected to NAD$^+$ or PBS treatment followed by canonical inflammasome activation did not exhibit any significant difference in IL-1β secretion or pyroptosis that was assessed by LDH release measurement, a marker for cell death (*Chan et al., 2013*; *Figure 2A*). To trigger the non-canonical inflammasome pathway, BMDMs were primed with Pam3CSK4, a TLR1/2 agonist, followed by cholera toxin B (CTB) and LPS (2 μg/ml) administration. The data showed that NAD$^+$ treatment resulted in a robust reduction of IL-1β release and cell death when compared to the PBS control group (*Figure 2A*). Furthermore, western blotting revealed that BMDMs cultured in the presence of NAD$^+$ exhibited a dramatic decrease of casp-11 expression and its downstream targets including casp-1, IL-1β, and cleaved GSDMD (*Figure 2B*). Moreover, we observed a prominent decrease in casp-1 expression under NAD$^+$ treatment that was constant over the time course of 16 hr. In contrast, BMDMs treated with PBS exhibited excessive casp-1 expression at 4 hr that was found to be absent after 16 hr (*Figure 2C*), which is consistent with the strong cytotoxicity leading to membrane permeabilization and release of casp-1 into the supernatant. Noteworthy, Pam3CK4-derived BMDM priming was not affected by NAD$^+$ since NF-κB as well as pro-caspase-1 levels had not been altered (*Figure 2A* and *Figure 2—figure supplement 1*) underlining the specific inhibition of casp-11. To visualize NAD$^+$-mediated blockade of pyroptotic macrophage death, BMDMs were treated with PBS or NAD$^+$, primed with Pam3CSK4, then stimulated with LPS and CTB, and cell viability and apoptosis were monitored using the IncuCyte live microscopy system. Hereby, we observed distinct longitudinal kinetics over 100 hr with complete disaggregation of cell integrity in the PBS group contrary to overall preserved cell structure in NAD$^+$-treated BMDMs (*Figure 2D*, *Figure 2—figure supplement 2*, *Video 1*). To rule out that NAD$^+$ impairs LPS internalization into cells, BMDMs were stimulated

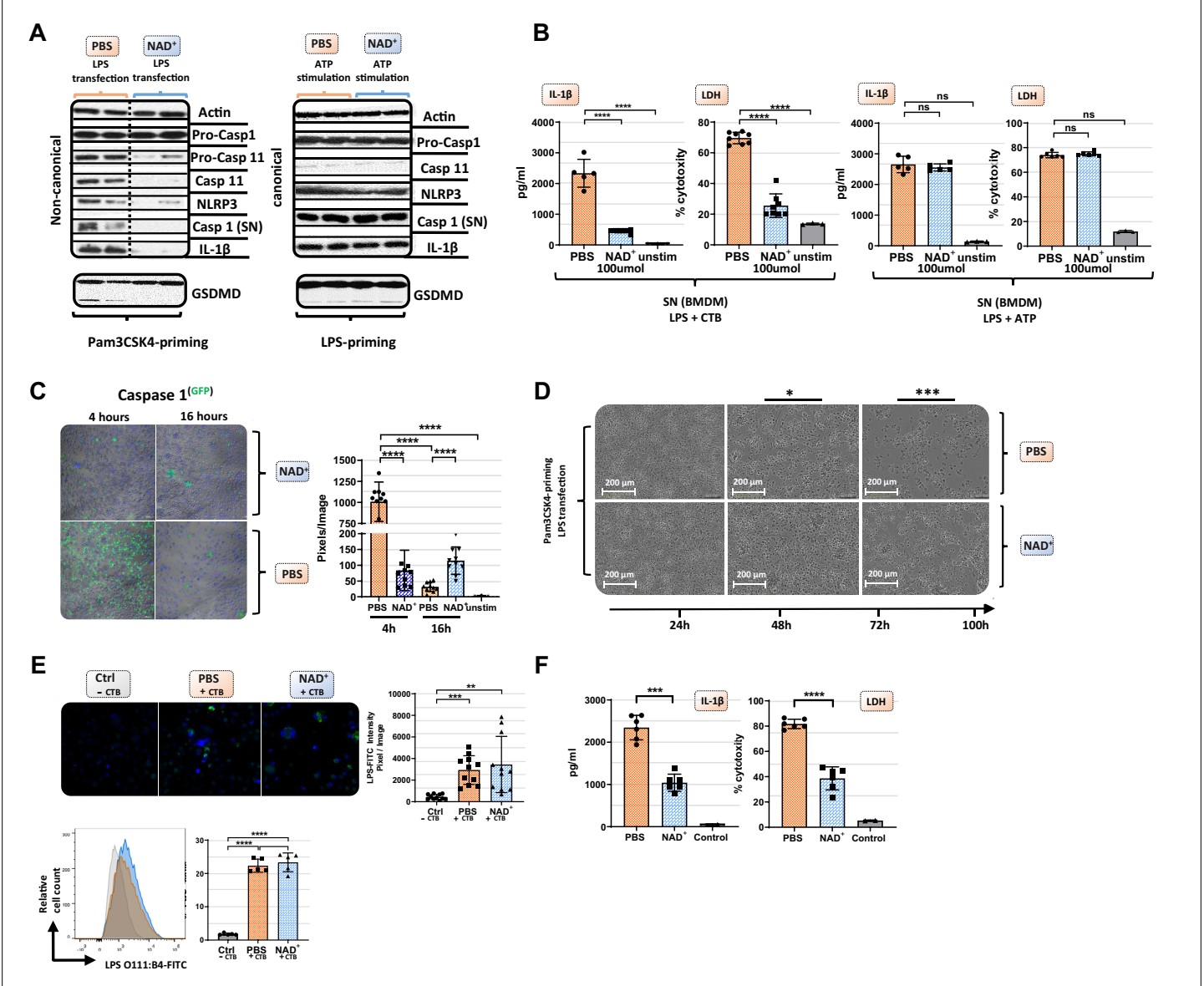

**Figure 2.** NAD⁺ specifically inhibits the non-canonical inflammasome by targeting caspase-11. Bone marrow was isolated from mice and bone marrow-derived macrophages (BMDMs) were differentiated in vitro. Subsequently, BMDMs were cultured in the presence of NAD⁺ or PBS. BMDMs were then primed with either Pam3CSK4 or lipopolysaccharide (LPS) O111:B4. Next primed BMDMs were stimulated with ATP or LPS and cholera toxin B (CTB). (**A**) Pro-casp-1, pro-casp-11, casp-11, NLRP3, casp-1, IL1β, and gasdermin D (GSDMD) expression were determined using western blot and (**B**) IL-1β secretion and LDH release were assessed in the supernatant. Column plots display mean with standard deviation (n=5-8). (**C**) Time-dependent caspase-1 expression was determined via active staining and assessed using a confocal microscope. Column plots display mean with standard deviation (n=5) (**D**) Cell viability and apoptosis were monitored using the IncuCyte live microscopy system. (**E**) LPS transfection with CTB was visualized by using FITC-coupled LPS and DAPI staining and quantified by confocal microscopy and flow cytometry. Column plots display mean with standard deiation (n=6) (**F**) For human experiments macrophages were differentiated from PBMC, primed with Pam3CSK4 and subsequently transfected with LPS and 0.25% Fugene HD Plus. Column plots display mean with standard deviation (n=6). Statistical significance was determined by using Student's t-test or one-way ANOVA. Asterisks indicate p-values *=p<0.05, **=p<0.01, and ***=p<0.001, only significant values are shown. All data depicted in this figure are provided as source data.

The online version of this article includes the following source data and figure supplement(s) for figure 2:

**Source data 1.** Raw data for *Figure 2A*: Original western blots.

**Source data 2.** Raw data for *Figure 2A*: Western blots with highlighted bands and sample labels.

**Source data 3.** Raw data for *Figure 2B*: ELISA mouse bone marrow-derived macrophages (BMDMs).

**Source data 4.** Raw data for *Figure 2C*: Caspase-1 staining.

*Figure 2 continued on next page*

*Figure 2 continued*

**Source data 5.** Raw data for *Figure 2D*: IncuCyte live microscopy.

**Source data 6.** Raw data for *Figure 2E*: Lipopolysaccharide (LPS) transfection staining.

**Source data 7.** Raw data for *Figure 2F*: ELISA human macrophages.

**Figure supplement 1.** NAD⁺ does not alter bone marrow-derived macrophage (BMDM)-derived NF-$\kappa$ B expression or phosphorylation.

**Figure supplement 1—source data 1.** Raw data for *Figure 2—figure supplement 1A*: Western blot.

**Figure supplement 1—source data 2.** Raw data for *Figure 2—figure supplement 1A*: Western blots bands with highlighted and sample labels.

**Figure supplement 1—source data 3.** Raw data for *Figure 2—figure supplement 1B*: Immunofluorescence.

**Figure supplement 2.** Unstimulated bone marrow-derived macrophage (BMDM) cell viability and apoptosis.

**Figure supplement 2—source data 1.** Raw data for *Figure 2—figure supplement 2*: IncuCyte live microscopy.

with CTB and LPS that was coupled to a fluorescent reporter (FITC) and transfection effectivity was assessed by fluorescence microscopy and flow cytometry. Our data indicated no significant difference between the PBS and NAD⁺-treated group (*Figure 2E*), suggesting that NAD⁺ does not alter LPS internalization. Notably, BMDMs only stimulated with LPS showed no internalization of LPS consistent with previous reports (*Kayagaki et al., 2013*). Casp-4 and -5 have been delineated as the human homolog of casp-11 in mice carrying out the same effector functions including pyroptosis induction and IL-1β secretion (*Shi et al., 2014*). As clinical relevance, we therefore tested whether NAD⁺ was also able to block the non-canonical pathway in human macrophages. Hence, human macrophages were differentiated from PBMC and treated with NAD⁺ followed by intracellular LPS transfection (Fugene) and IL-1β secretion and cytotoxicity were quantified. The results indicated that NAD⁺ treatment significantly dampened both IL-1β secretion and pyroptosis (*Figure 2F*), underscoring its therapeutic potential. Collectively, our results suggest that NAD⁺ acts directly on macrophages by targeting specifically the non-canonical inflammasome signaling machinery.

## NAD⁺ inhibits the non-canonical inflammasome via STAT-1/IFN-β pathway blockade

Although our data emphasized that NAD⁺ blocks the non-canonical inflammasome pathway, the underlying mechanisms remained yet to be determined. Therefore, we performed RNA-sequencing of Pam3CSK4 primed BMDMs that were treated with PBS or NAD⁺ and subsequently stimulated with CTB+LPS O111:B4. Interestingly, when blotting gene expression differences in a Venn diagram, we found strikingly more genes commonly expressed in the NAD⁺ and control group when compared to the PBS-treated group (*Figure 3A*). Gene ontology enrichment analysis revealed a significant downregulation of genes involved in the antiviral response in addition to the cellular response to the type I IFN, IFN-β, when comparing NAD⁺ and PBS-treated groups (*Figure 3B*). Type I IFN are known to promote the expression of over 2000 IFN-stimulated genes (ISGs), translated into ISGs-induced proteins which have been shown to act by enhancing pathogen detection and restrict their replication (*Schneider et al., 2014*). Recently, it was reported that type I IFNs are required for casp-11 expression contributing to non-canonical inflammasome activation (*Rathinam et al., 2012*; *Tang et al., 2018*). Consistently, LPS-stimulated macrophages from TRIF-deficient mice displayed impaired casp-11 expression, implying a context-dependent role for type I IFN in the regulation of caspase-11 activity (*Rathinam et al., 2012*). Indeed, when comparing expression of genes involved in IFN-β signaling through cluster analysis we found a significant decrease in a broad range in genes in the NAD⁺-treated group (*Figure 3C*). Most strikingly, GTPases and guanylate binding proteins involved in the downstream signaling of IFN-β were significantly downregulated (*Figure 3C* and *Figure 3D*) while IFN-β-receptor 2 expression remained unaffected (*Figure 3C*). Recently, IFN-inducible GTPases and

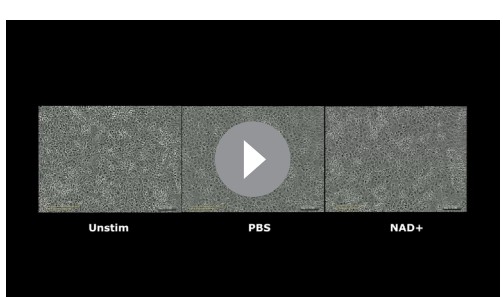

**Video 1.** Live Microscopy of NAD+ and PBS treated BMDMs subjected to non-canonical inflammasome activation.

https://elifesciences.org/articles/88686/figures#video1

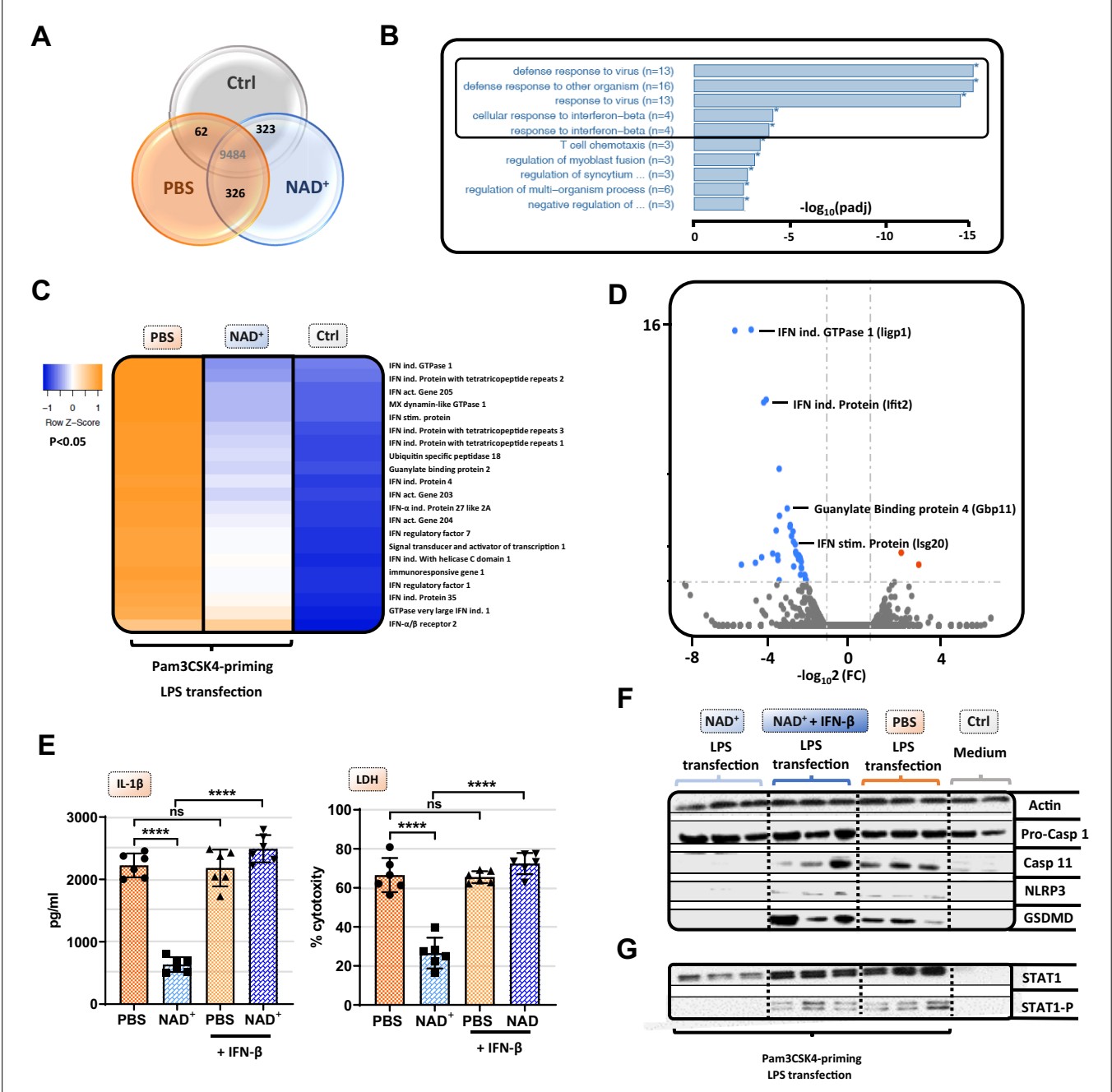

**Figure 3.** NAD+-mediated inhibition of the non-canonical inflammasome is based on an impaired response to IFN-β. Differentiated bone marrow-derived macrophages (BMDMs) were cultured in the presence of NAD+ or PBS. BMDMs were then primed with Pam3CSK4, subsequently stimulated with lipopolysaccharide (LPS) and cholera toxin B (CTB) and RNA-sequencing was performed. Unstimulated BMDMs served as controls. (**A**) Venn diagram plotting common gene expression between all three groups. (**B**) Gene ontology enrichment analysis displaying the highest significant pathways differing when comparing NAD+ and PBS-treated BMDMs. (**C**) Expression cluster analysis of genes involved in IFN-β signaling through cluster analysis depicted in a heat map. (**D**) Volcano plot displaying the most significant genes up- or downregulated comparing NAD+ and PBS-treated BMDMs. (**E**) Stimulated BMDMs were additionally treated with recombinant INF-β, and IL-1β and LDH release were measured. Column plots display mean with standard deviation (n=6) (**F**) Moreover, pro-casp-1, casp-11, NLRP3, gasdermin D (GSDMD), (**G**) signal transducer activator of transcription-1 (STAT-1), and phospho-STAT-1 expression were assessed by western blot. Statistical significance was determined by using Student's t-test or one-way ANOVA. Asterisks indicate p-values *=p<0.05, **=p<0.01, and ***=p<0.001, only significant values are shown. All data depicted in this figure are provided as source data.

The online version of this article includes the following source data for figure 3:

*Figure 3 continued on next page*

*Figure 3 continued*

**Source data 1.** Raw data for *Figure 3E*: ELISA bone marrow-derived macrophage (BMDM).

**Source data 2.** Raw data for *Figure 3F*: Original western blots.

**Source data 3.** Raw data for *Figure 3F*: Western blots with highlighted bands and sample labels.

**Source data 4.** Raw data for *Figure 3G*: Original western blots.

**Source data 5.** Raw data for *Figure 3G*: Western blots with highlighted bands and sample labels.

guanylate binding proteins have been assigned a crucial role for the intracellular recognition of LPS and linked caspase-11 activation (*Tang et al., 2018*; *Pilla et al., 2014*). Thus, to test if NAD$^+$ mediated non-canonical inflammasome blockade via IFN-β, NAD$^+$ or PBS-treated BMDMs were primed with Pam3CSK4 and subsequently stimulated with LPS O111:B4+CTB and 1000 U/ml of recombinant IFN-β. Strikingly, administration of recombinant IFN-β resulted in a complete reversal of NAD$^+$-mediated blockade of IL-1β secretion and pyroptosis (*Figure 3E*). Moreover, IFN-β administration restored casp-11, NLRP3, and GSDMD expression in the NAD$^+$-treated group (*Figure 3F*). It is well established that signal transducer activator of transcription-1 (STAT-1) phosphorylation constitutes the link between intracellular type I IFN signaling and the transcription of ISGs through nuclear translocation (*Stark and Darnell, 2012*; *Ivashkiv and Donlin, 2014*). Notably, our RNA-sequencing data indicated a significant downregulation of STAT-1 (*Figure 3C*). Moreover, we have previously shown that NAD$^+$ administration dampens the expression and activation of transcription factors such as STAT-5 (*Elkhal et al., 2016*). To test, whether NAD$^+$ blocks IFN-β signaling via STAT-1, BMDMs were subjected to NAD$^+$ or PBS followed by non-canonical inflammasome stimulation and recombinant IFN-β. After 16 hr STAT-1 expression and phosphorylation were assessed by western blotting. Consistent with our previous results, NAD$^+$ treatment downregulated expression levels of STAT-1 and phospho-STAT-1. In contrast, addition of recombinant IFN-β treatment to NAD$^+$-treated BMDMs restored STAT-1 and phospho-STAT-1 expression that was equivalent to the PBS-treated group (*Figure 3G*). Taken together, our data indicate that NAD$^+$ impedes non-canonical inflammasome activation via IFN-β/STAT-1 blockade (*Figure 4*).

## NAD$^+$ increases caspase-1 KO mice resistance to endotoxic shock via systemic IL-10 production

Caspase-11 KO mice have been reported to be resistant toward lethal doses of LPS inducing septic shock (*Kayagaki et al., 2013*). However, upon priming with TLR3 instead of a TLR4 ligand, casp-11 KO mice merely exhibit partial resistance toward LPS-induced shock with a 50–60% survival rate (*Kayagaki et al., 2013*; *Hagar et al., 2017*). Our data indicate that NAD$^+$ prevents LPS-induced cell death via the non-canonical inflammasome pathway and casp-11 blockade. We thus tested whether NAD$^+$ could achieve similar protection against septic shock in WT vs casp-11 KO mice. Casp-11 KO mice were intraperitoneally injected with NAD$^+$ and PBS and treated with 6 mg/kg poly(I:C) 6 hr prior to LPS administration. Consistent with previous studies the results indicated a modest resistance of casp-11 KO mice (40% survival). In high contrast, both WT and casp-11 KO mice subjected to NAD$^+$ exhibited 85–100% survival, respectively, when compared to casp-11 KO mice that were treated with PBS, suggesting the existence of an alternative protective pathway against septic shock that is casp-11 independent. WT mice, treated with 6 mg/kg poly(I:C) followed by LPS (54 mg/kg) administration, not only survived but fully recovered 7 days later, underscoring the unique and robust therapeutic effect of NAD$^+$ in septic shock. Previous studies have reported inferior outcomes of IL-10-/- mice in septic shock (*Latifi et al., 2002*; *Berg et al., 1995*) pointing out a 20-fold lower lethal dose of LPS compared to WT mice (*Berg et al., 1995*). Moreover, IL-10 itself has been shown to prevent mice from septic shock-induced death after a single administration (*Howard et al., 1993*). We have previously delineated immunosuppressive properties of NAD$^+$ via a systemic production of IL-10, a robust immunosuppressive cytokine. In addition, we have described the pivotal role of NAD$^+$ protecting toward EAE and allograft rejection via an increased frequency of IL-10 producing CD4$^+$ T cells (*Tullius et al., 2014*; *Elkhal et al., 2016*). To test if IL-10 plays an additional protective role in the context of NAD$^+$-mediated protection toward LPS-induced death, WT mice treated with NAD$^+$ or PBS subjected to intraperitoneal LPS injection (54 mg/kg) and IL-10 expression by macrophages, dendritic cells, and T cells was assessed 15 hr after LPS administration. Consistent with our previous studies (*Tullius*

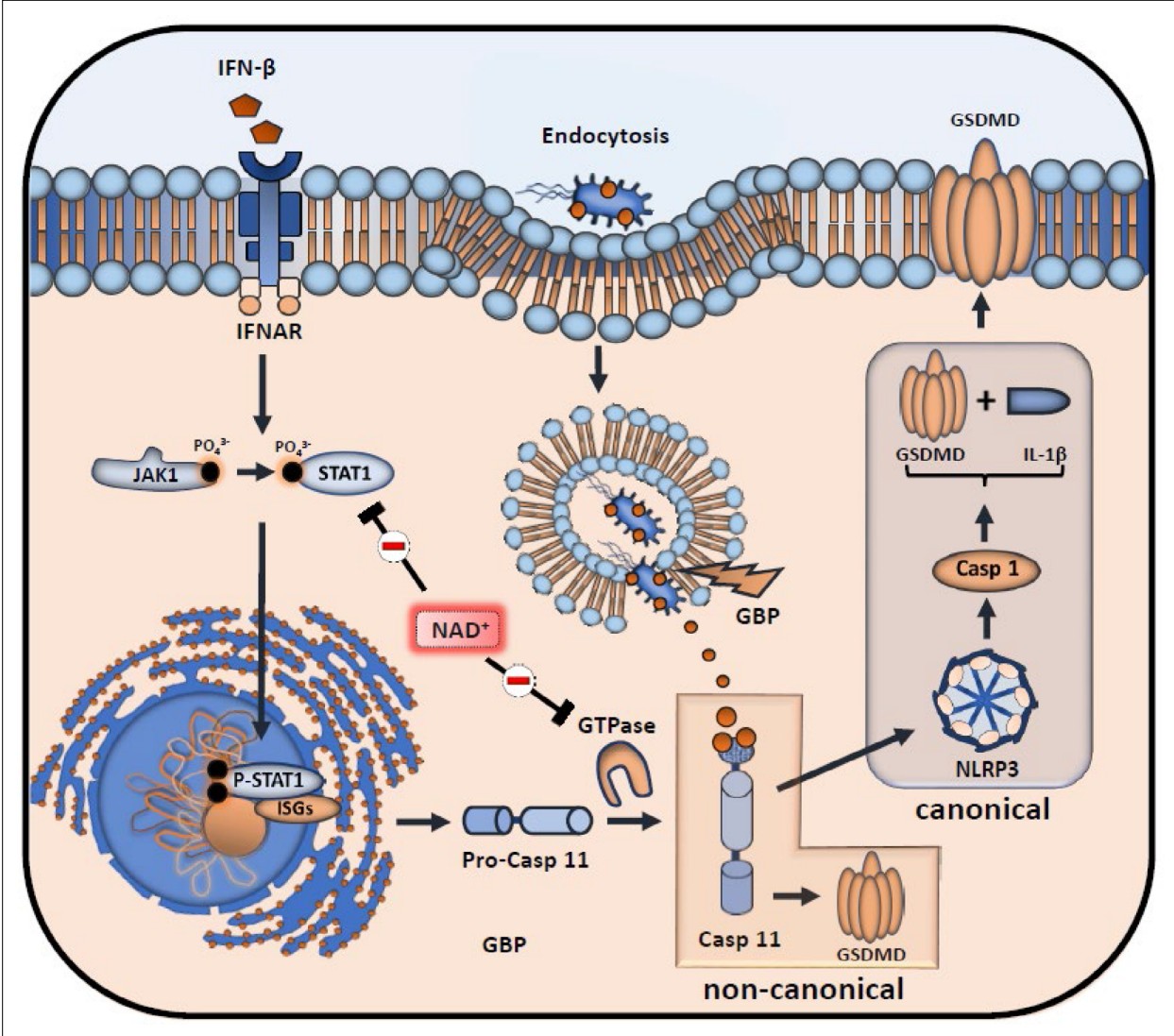

**Figure 4.** Inhibitory effects of NAD$^+$ on IFN-β downstream signaling and inflammasome activation. NAD$^+$ inhibits signal transducer activator of transcription-1 (STAT-1) expression and phosphorylation, thus compromising the intracellular response to IFN-β. Subsequently, stimulation of the IFNAR receptor by IFN-β leads to a decreased transcription of pro-caspase-11 as well as IFN-stimulated genes (ISGs) (IFN-inducible GTPases and GBPs). Due to diminished caspase-11 levels, non-canonical inflammasome activation through intracellular, gram-negative bacteria opsonization by GBPs is significantly inhibited.

et al., 2014; *Elkhal et al., 2016*), we found significantly augmented frequencies of IL-10 producing CD4$^+$ and CD8$^+$ T cells (*Figure 5C*). Moreover, we detected a dramatic increase of IL-10 production by macrophages, but not the DC population (*Figure 5B*). Interestingly, IL-10 has been described to inhibit macrophage function and proinflammatory cytokine production in both, human (*de Waal Malefyt et al., 1991*) and mice (*Fiorentino et al., 1991*). Moreover, autocrine IL-10 secretion of macrophages was found to decrease pro-IL-1β concentration by promoting STAT-3 expression (*Sun et al., 2019*). To investigate the potential autocrine impact of an augmented IL-10 production on macrophage self-regulation, we administered combined IL-10 neutralizing antibody and IL-10 receptor antagonist to BMDMs primed with Pam3CSK4 and stimulated with CTB and LPS O111:B4. The results showed that neutralization of the autocrine IL-10 signaling pathway dampened NAD$^+$-mediated decrease of IL-1β secretion and reversed pyroptotic cell death partially (*Figure 5D*). To further investigate the relevance of our in vitro findings, IL-10$^{-/-}$ mice were treated with NAD$^+$ or PBS, subjected to LPS (54 mg/kg) and survival was monitored. Consistent with previous reports (*Latifi et al., 2002*; *Berg et al., 1995*), mice lacking IL-10 exhibited an inferior protection against septic shock when compared to WT animals.

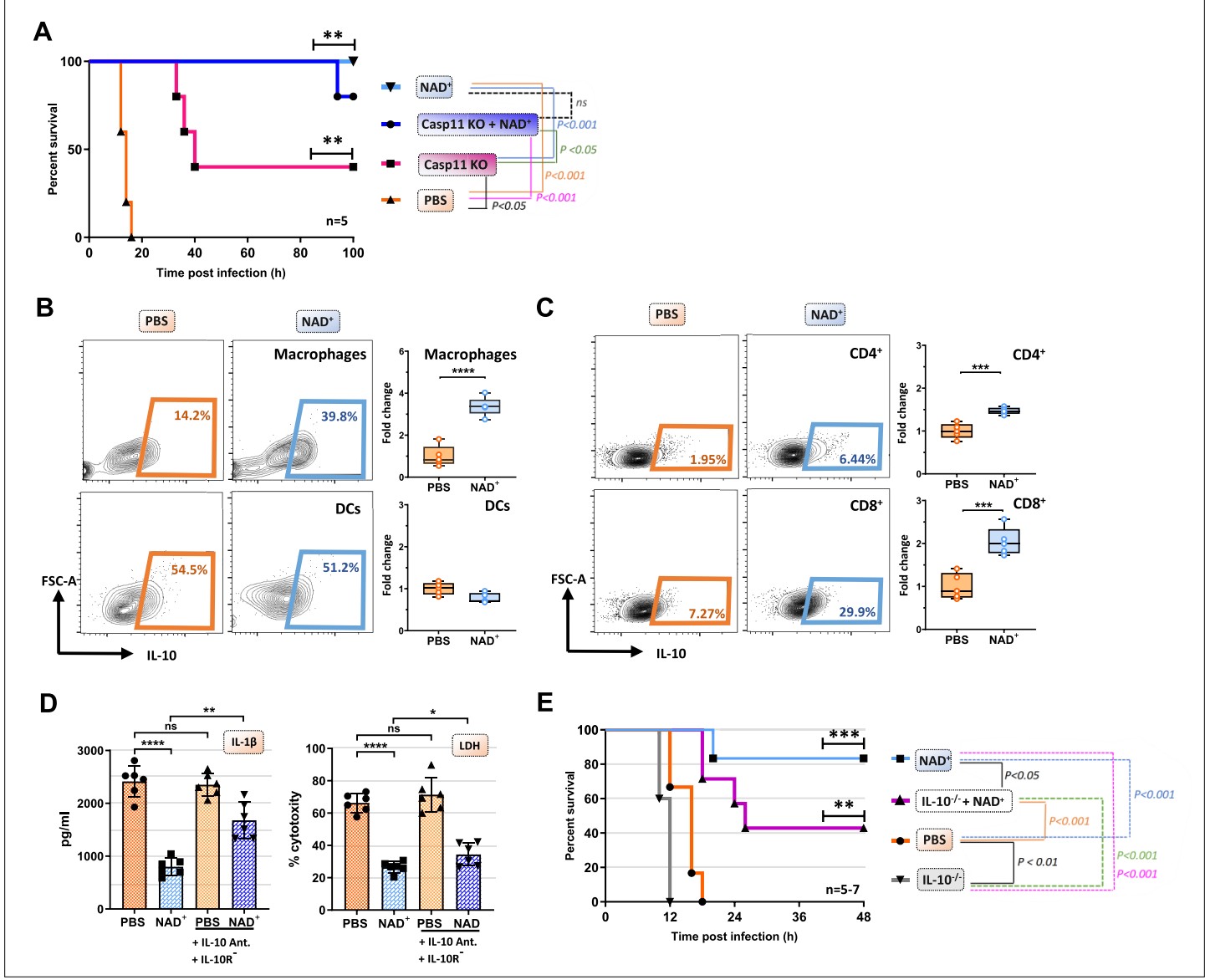

**Figure 5.** IL-10 constitutes an additional mechanism mediating the protective capacities of NAD+ in the context of septic shock. (**A**) Caspase-11 KO (knockout) mice were treated with NAD+ or PBS. Subsequently mice were subjected to poly(I:C) prior to lipopolysaccharide (LPS) injection and survival was monitored (n=5, 2 independent survival experiments). Mice treated with either NAD+ or PBS were injected with LPS and after 10 hr, splenic frequencies of IL-10 producing (**B**) macrophges and dendritic cells (**C**) and CD4+ and CD8+ T cells were assessed by flow cytometry. Box plots display fold change of leukocyte proportions as mean with standard deviation (n=5) (**D**) Bone marrow-derived macrophages (BMDMs) treated with NAD+ or PBS were stimulated with LPS and cholera toxin B (CTB) in the presence of IL-10 neutralizing antibodies and IL-10 receptor antagonists. Subsequently IL-1β and LDH release were assessed. Column plots display mean with standard deviation (n=6) (**E**) IL-10-/- mice treated with NAD+ or PBS were challenged with LPS and survival was monitored (n=5–7, 2 independent survival experiments). Statistical significance was determined by using Student's t-test or one-way ANOVA while survival data were compared using log-rank Mantel-Cox test. Asterisks indicate p-values *=p<0.05, **=p<0.01, and ***=p<0.001, only significant values are shown. All data depicted in this figure are provided as source data.

The online version of this article includes the following source data for figure 5:

**Source data 1.** Raw data for *Figure 5A*: Casp11 knockout (KO) survival.

**Source data 2.** Raw data for *Figure 5B*: FACS macrophages and DCs.

**Source data 3.** Raw data for *Figure 5C*: FACS CD4+ and CD8+ T cells.

**Source data 4.** Raw data for *Figure 5D*: ELISA bone marrow-derived macrophage (BMDM).

**Source data 5.** Raw data for *Figure 5E*: IL-10-/- survival.

More importantly, IL-10$^{-/-}$ mice subjected to NAD$^+$ exhibited a compromised survival (*Figure 5E*), suggesting that systemic production of IL-10 following NAD$^+$ administration plays a pivotal role in NAD$^+$-mediated protection against septic shock.

## Discussion

Previously, we have delineated the protective role of NAD$^+$ in the context of L. m. infection, a gram-positive bacterium (*Rodriguez Cetina Biefer et al., 2018*). However, it remained unclear whether NAD$^+$ conveyed resistance toward L. m. by an augmented bacterial clearance or rather through its immunosuppressive effects dampening pathological systemic inflammation. Although the cell membrane of L. m. has been shown to bear lipoteichoic acids, which resemble the endotoxin LPS from gram-negative bacteria in both, structure and function, it is widely considered as an intracellular bacterium (*Farber and Peterkin, 1991*). In our current study, we administered a lethal dose of pathogenic *E. coli*, that is well known to promote septic shock, and showed that NAD$^+$ also protected toward a lethal dose of this gram-negative bacterium. More importantly, we demonstrate that NAD$^+$ conveys protection toward septic shock by specifically inhibiting the non-canonical inflammasome but not via bacterial clearance. Mechanistically, NAD$^+$ impedes pro-casp-11 and casp-11 expression in macrophages blocking non-canonical-derived GSDMD cleavage and NLRP3 inflammasome activation, thus inhibiting pyroptotic cell death and proinflammatory cytokine release. The resistance of NAD$^+$-treated WT mice against *E. coli* and LPS-induced septic shock reflected the robust inhibitory effect observed in vitro of NAD$^+$ on the non-canonical inflammasome signaling machinery.

Until now, the exact mechanism how pro-casp-11 expression and its maturation to casp-11 is regulated remains unclear. Given the low basal expression of both pro-casp-11 and casp-11 (*Schauvliege et al., 2002*), a priming signal is required for initiating the non-canonical inflammasome pathway and macrophage sensing of intracellular LPS (*Yang et al., 2015b*). Previous work has demonstrated that transcriptional induction of the pro-casp-11 isoforms p42 and p38 in macrophages requires type I IFN stimulation (*Schauvliege et al., 2002*; *Yen and Ganea, 2009*) while IFN-β has been shown to promote transcriptional induction and processing of caspase-11 (*Rathinam et al., 2012*). In line with these findings, CTB treatment of macrophages primed with Pam3csk4 failed to elicit IL1-β release compared to LPS primed controls while exogenous administration of IFN-β in turn restored CTB-induced IL-1β production (*Rathinam et al., 2012*) underscoring the transcriptional role of type I IFN. Our RNA-sequencing results indicated a dampened cellular response toward IFN-β while western blotting revealed a significant downregulation of both, pro-casp-11 and casp-11, suggesting a transcriptional downregulation of both enzymes. Consistently, NAD$^+$ decreased STAT-1 expression and phosphorylation, which constitutes the mechanistic link between extracellular type I IFN stimulation and transcriptional effects through translocation of phosphorylated STAT-1 to the nucleus inducing ISGs (*Ivashkiv and Donlin, 2014*). Thus, treatment of stimulated macrophages subjected to NAD$^+$ with recombinant IFN-β restored STAT-1 signaling, caspase-11 expression, and GSDM cleavage which translated into reconstituted IL-1β production and LDH release. Collectively, NAD$^+$ mitigates the intracellular response to IFN-β that leads to non-canonical inflammasome induction by suppressing macrophage-derived STAT-1 expression and phosphorylation. Furthermore, we showed that NAD$^+$ treatment improved resistance of casp-11 KO mice toward poly I:C primed septic shock. More importantly, WT mice treated with NAD$^+$ exhibited 100% survival while casp-11 KO mice treated with PBS exhibited a modest 40% survival, suggesting that NAD$^+$ promotes survival beyond non-canonical inflammasome blockade. Our previous studies have delineated the effects of NAD$^+$ on various immune cells such as dendritic cells and CD4$^+$ T cells including Th1, Th17, regulatory type 1 (Tr1), and Treg cells communicated exclusively through MCs (*Tullius et al., 2014*; *Elkhal et al., 2016*; *Rodriguez Cetina Biefer et al., 2018*). Thereby, NAD$^+$ treatment promoted MC-derived induction of TR1 cells that resulted into increased systemic levels of IL-10. Latter one was found to play a cardinal feature during bacterial infection as MC$^{-/-}$ mice were more susceptible to L. m. infection than WT animals when treated with NAD$^+$. Here, we found a direct effect of NAD$^+$ on macrophages by specifically inhibiting the non-canonical inflammasome and promoting IL-10 production. Polymorphisms in the IL-10 locus or IL-10R deficiencies have been linked to severe intestinal inflammatory diseases in both, human and mice (*Franke et al., 2008*; *Franke et al., 2010*; *Kühn et al., 1993*; *Begue et al., 2011*). More importantly, mice deficient for IL-10 have been shown to display elevated inflammasome activation and IL-1β production resulting in severe colitis (*Zhang et al., 2014*) or Ag-induced

arthritis (*Greenhill et al., 2014*). When inhibiting the autocrine pathway for IL-10 through combined receptor antagonization and IL-10 neutralization, we found a pronounced increase of IL-1β production of NAD$^+$-treated macrophages stimulated with CTB and LPS (*Figure 4D*). This is consistent with previous reports showing that autocrine IL-10 signaling interferes with the transcription of pro-IL-1β (*Sun et al., 2019*). LDH release, in turn, was only restored partly possibly due to missing effects of second party leucocytes secreting IL-10 in vivo such as Tr1 cells which have been shown to inhibit the transcription of IL-1β and inflammasome-mediated activation of caspase-1 (*Yao et al., 2015*). More recently, casp-8, that plays a central role in apoptosis, has been reported as an important mediator of endotoxemia resistance and LPS-driven systemic inflammation. Since our RNA-sequencing results revealed a dramatically attenuated cellular response toward type I IFN with downregulation of various interferon regulatory factors, that have been reported as major regulators of casp-8 (*Apelbaum et al., 2013*; *Newton et al., 2019*), it is possible that NAD$^+$ may exert protection against septic shock by altering caspase-8 expression as well. Although we have previously reported the protective effect of NAD$^+$ against apoptosis of activated CD4$^+$ T cells (*Tullius et al., 2014*), it remains yet to be determined how NAD$^+$ impacts executioner proteins of other cell death processes such as apoptosis and necroptosis.

Notably, both casp-8 and casp-11 have been found dispensable in the hematopoietic compartment that produces the proinflammatory cytokines necessary to initiate shock (*Mandal et al., 2018*). Thus, NAD$^+$ treatment may improve resistance of casp-11 KO mice to septic shock by also dampening the initiating proinflammatory cytokine cascade via its systemic IL-10 cytokine production. Importantly, while inhibiting macrophage-derived inflammasome function, NAD$^+$ does not interfere with NF-κB signaling which has been shown to promote various inflammatory and autoimmune diseases when dysregulated (*Liu et al., 2017*). Taken together, we dissected the dichotomous capacity of NAD$^+$ to dampen auto- and allo-immunity while concomitantly protecting toward severe bacterial infection, outlining its unique effects in the context of septic shock.

## Materials and methods
### Animals
Young (8–12 weeks) C57BL/6, B6.129P2-IL10tm1Cgn/J, and B6.129S4(D2)-Casp4tm1Yuan/J mice were purchased from Charles River Laboratory, Wilmington, MA, USA. All mice were male, age-matched and experimental and control animals were housed separately. Animals and samples were randomly assigned to either the control or treatment group to ensure biological diversity. The study protocol was approved by the Brigham and Women's Hospital Institutional Animal Care and use Committee (IACUC) (animal protocol #2018N000049). All mice were male, age-matched and experimental and control animals were housed separately. Owing to the exploratory nature of our study, we did not use randomization and blinding. No statistical methods were used to predetermine sample size. All animals were maintained in specific pathogen-free conditions at the Brigham and Women's Hospital animal facility in accordance with federal, state, and institutional guidelines. Animals were maintained on 12 hr light, 12 hr dark cycle in facilities with an ambient temperature of 19–22°C and 40–60% humidity and were allowed free access to water and standard chow. Euthanasia was performed by cervical dislocation following anesthesia with isoflurane (Patterson Veterinary, Devens, MA, USA).

### Murine bone marrow-derived macrophage differentiation and culture
8- to 12-week-old C57BL/6 mice were euthanized by cervical dislocation, sprayed with alcohol and skin was removed to expose femurs. The femur was flushed with ice-cold PBS and the obtained bone marrow was filtered through 70 µm Nylon cell strainer. After washing with PBS, red blood cell lysis was performed using ammonium-chloride-potassium solution (Fisher Scientific) and the reaction was blocked with complete Dulbecco's modified eagle medium (DMEM) (Fisher Scientific) supplemented with 10% endotoxin-free bovine serum and PS. To minimize fibroblast contamination cells were cultured in complete DMEM at 37°C, 5% CO$_2$ and non-adherent cells were collected after 30 min. Bone marrow cells were then differentiated into macrophages in DMEM supplemented with 10% endotoxin-free bovine serum, PS, and 40 ng/ml murine GM-CSF (Abcam) for 8 days. Medium was changed every 2 days to remove non-adherent cells.

## Canonical and non-canonical inflammasome activation in murine macrophages

After 8 days of culture the medium was replaced by 40 ng/ml GM-CSF containing 100 µmol NAD$^+$ culture medium. For 2 following days NAD$^+$ was added daily until stimulation. To induce canonical and non-canonical inflammasome activation in murine macrophages, NAD$^+$-treated and control BMDMs were cultured overnight in a 24-well plate at $1\times10^6$ cells/ml and afterward primed with 1 µg/ml Pam3CSK4 or 1 µg/ml LPS O111:B4 (Sigma) for 5–6 hr. Primed BMDMs were then stimulated for 16 hr with either 5 mmol ATP (canonical inflammasome stimulation) or 2 µg/ml LPS O111:B4 and 20 µg/ml CTB (Sigma) to allow LPS entry (non-canonical inflammasome stimulation) where indicated. To test the effect of NAD$^+$ on type 1 IFN and STAT1 signaling, BMDMs were cultured overnight in a 24-well plate at $1\times10^6$ cells/ml and afterward primed with 1 µg/ml Pam3CSK4 or 1 µg/ml LPS O111:B4 (Sigma) for 5–6 hr. Primed BMDMs were then stimulated for 16 hr 2 µg/ml LPS O111:B4, 20 µg/ml CTB, and U/ml recombinant IFN-β.

## ELISA

Expression of macrophage-derived murine IL-1β, IL-18, and human IL-1β was analyzed in the cell culture supernatant by commercial ELISA kits (Invitrogen) following the manufacturer's recommended procedures.

## Pyroptosis assay

Pyroptotic cell death was measured by assessing LDH release in the cell culture supernatant of human and murine macrophages using a CytoTox 96 Non-radioactive Cytotoxic Assay (Promega) following the manufacturer's recommended procedures.

## RNA extraction and RNA-sequencing

BMDMs were harvested and differentiated as outlined in the particular section. After 8 days of culture the medium was replaced by 40 ng/ml GM-CSF containing culture medium (control group) or 40 ng/ml GM-CSF containing 100 µmol NAD$^+$ culture medium (NAD$^+$-treated group). For 2 following days NAD$^+$ was added daily. Subsequently, NAD$^+$-treated and control BMDMs were cultured overnight in a 24-well plate at $1\times10^6$ cells/ml and afterward primed with 1 µg/ml Pam3CSK4 or 1 µg/ml LPS O111:B4 (Sigma) for 5–6 hr. Primed BMDMs were then stimulated for 16 hr with 2 µg/ml LPS O111:B4 and 20 µg/ml CTB (Sigma) to allow LPS entry. Another set of BMDMs were differentiated without additional treatment serving as naïve controls. Subsequently, RNA was extracted with the RNAqueous extraction kit (Applied Biosystems), according to the manufacturer's protocols. Briefly, cells were homogenized in lysis buffer (total volume of 0.5 ml) and passed through a column. After successive washes, RNA was eluted. RNA-sequencing was commercially performed by Novogene Co., Ltd. In brief, mRNA was enriched from total RNA using oligo(dT) beads and subsequently fragmented randomly in fragmentation buffer, followed by cDNA synthesis using random hexamers and reverse transcriptase. After first-strand synthesis, a custom second-strand synthesis buffer (Illumina) was added with dNTPs, RNase H, and *E. coli* polymerase I to generate the second strand by nick-translation. The final cDNA library is ready after a round of purification, terminal repair, A-tailing, ligation of sequencing adapters, size selection, and PCR enrichment.

## Isolation and differentiation of human macrophages from PBMCs

Blood was obtained from healthy adult volunteers with the only purpose to isolate PBMCs in order to create a basis for macrophage cultures. Blood withdrawal was performed in accordance with the guidelines of and approved by the Institutional Review Board of the Charité Universitätsmedizin Berlin (EA4/006/22). Informed consent and consent to publish was obtained from each volunteer in accordance with the Declaration of Helsinki. All personal information collected from volunteers were treated with utmost confidentiality. For experiments on human macrophages, PBMCs were isolated by performing a density centrifugation in SepMate tubes (StemCell) using lymphoprep (StemCell) density gradient medium. PBMCs were then plated in DMEM culture medium supplemented with standard antibiotics, 10% FCS, and human 50 ng/ml GM-CSF (PeproTech) at a density of $1\times10^6$ cells/ml. The medium was changed every 2–3 days until the cells reached a full confluence.

### Non-canonical inflammasome induction in human macrophages

After 8 days of culture the medium was replaced by 50 ng/ml GM-CSF containing 100 µmol $NAD^+$ culture medium. For 2 following days $NAD^+$ was added daily until stimulation. To induce non-canonical inflammasome activation in human macrophages, cells were primed with 1 µg/ml Pam3CSK4 for 5–6 hr. Subsequently, the medium was replaced, and cells were treated with 3 µg/ml LPS O111:B4 and 0.25% (vol/vol) Fugene HD Plus (Promega) to cause transfection. Finally, plates were centrifuged at $805 \times g$ for 2 min and subsequently cultured for 20 hr at 37°C, 5% $CO_2$.

### Western blot

For western blot analysis, proteins were extracted using RIPA buffer and the concentrations determined using Pierce BCA Protein Assay Kit. Subsequently, proteins were resolved in SDS-PAGE, transferred to 0.45 µm nitrocellulose membranes (Bio-Rad), blocked with 5% non-fat dry milk in PBS with 0.1% Tween 20, and processed for immunodetection. The following primary antibodies were used according to the manufacturer's instructions: pro-caspase-1 (#ab179515, Abcam), caspase-1 (#14-9832-82, eBioscience), IL-1β (AF-401-NA, R&D Systems), NLRP3 (#768319, R&D Systems), caspase-11 (#mab8648, R&D Systems), GSDMD (ab209845, Abcam), P-STAT-1 (#9167S, Cell Signaling), STAT-1 (#9172S, Cell Signaling), NF-κB-p65 (#49445S, Cell Signaling), NF-κB-p52 (#4882S, Cell Signaling), β-actin (ab3280, Abcam). Antibody detection was performed with HRP-coupled goat secondary anti-mouse or anti-rabbit antibodies (ImmunoResearch), followed by ECL reaction (Perkin Elmer) and exposure to Fuji X-ray films. Finally, films were scanned, and signals quantified using the web-based image processing software ImageJ (NIH).

### Analysis of LPS transfection efficiency

For intracellular detection of LPS, primed BMDMs were stimulated with 20 µg/ml CTB and FITC-conjugated LPS O111:B4 for 16 hr, washed twice with PBS, fixed in 4% PFA containing PBS and DAPI for 10 min, and subsequently analyzed using a confocal microscope and flow cytometry. To determine transfection efficiency using confocal microscopy, FITC-stained pixels per image were quantified using the web-based image processing software ImageJ (NIH).

### Caspase-1 assay

To determine the expression of caspase-1, primed BMDMs were stimulated with 20 µg/ml CTB and 2 µg/ml LPS O111:B4 for 4 and 16 hr, respectively, washed twice with PBS, stained using a caspase-1 active staining kit (Abcam) including caspase-1 staining (fluorescent green) and DAPI staining (fluorescent blue) according to the manufacturer's protocol and analyzed using a confocal microscope.

### Endotoxic shock model

8- to 12-week-old C57BL/6 mice were treated with 40 mg $NAD^+$ for 2 following days before intraperitoneal injection of 54 mg/kg LPS O111:B4 or LPS O55:B5. Where indicated mice were administered 6 mg/kg poly(I:C) 6 hr prior to LPS administration. Consequently, survival and body temperature were monitored every 2–4 hr for up to 100 hr. To assess the amount of systemic IL-1β and IL-18 by ELISA (both Invitrogen), mice were euthanized by decapitation 10 and 15 hr after LPS injection serum was isolated from collected blood.

### Flow cytometric analysis

To analyze splenic lymphocytes for the intracellular expression of IL-10, mice were challenged with 54 mg/kg LPS O111:B4 for 10 hr and euthanized by cervical dislocation subsequently. Spleens were harvested in a sterile environment and single-cell suspensions were obtained using a 70 µm Nylon cell strainer. Then, 1×106 splenocytes per animal per condition were cultured in RPMI 1640 (#10-040-CV, Corning) supplemented with 10% BenchMark Fetal Bovine Serum (#100-106, Gemini), 1% penicillin/streptomycin (#30-002 CI, Corning), 2 mM L-glutamine (#25-005 CI, Corning), 20 ng/ml phorbol 12-myristate 13-acetate (#P8139-1MG, Sigma-Aldrich), 1 µg/ml ionomycin (#I9657-1MG, Sigma-Aldrich), and 0.67 µl/mL BD GolgiStop (#554724, BD Biosciences) for 4 hr at 37°C and 5% $CO_2$ in 1 ml-volumes in a 12-well plate. After 4 hr, the cells were collected from each 12-well plate well and prepared for flow cytometry by staining the surface epitopes in flow staining buffer consisting of 1× DPBS supplemented with 1.0% (wt/vol) bovine serum albumin (#A2153, Sigma-Aldrich) and 0.020%

sodium azide (#S8032, Sigma-Aldrich) for 25 min at 4°C. Then, the cells were fixed and permeabilized with the eBioscience Foxp3 Fixation/Permeabilization concentrate and diluent cocktail (#00-5523-00, Invitrogen) for 30 min at 4°C. Finally, the intracellular cytokine target was stained in 1× permeabilization buffer diluted from 10× eBioscience Foxp3 Permeabilization Buffer (#00-5523-00, Invitrogen) with deionized water. Finally, the stained samples were analyzed on a FACS Canto II (BD Biosciences, San Jose, CA, USA) flow cytometer, and the resultant flow cytometry standard (FCS) files were analyzed with FlowJo version 10 (FlowJo LLC, Ashland, OR, USA).

## Bacterial infection model

Frozen stock suspensions of *E. coli* (Migul) (ATCC, 700928) were obtained from ATCC and cultured in 5 ml Luria-Bertani medium at 37°C. Bacterial concentration was determined by plating 100 µl, 10-fold serial diluted bacterial samples and counting the colony-forming units (CFU) after overnight incubation at 37°C. One day prior to injection 1 ml of culture was reinoculated into 5 ml of medium and incubated for 3–5 hr using a 37°C shaker at 250 rpm agitation. Bacterial cultures were then centrifuged for 10 min at 3000 rpm and washed twice with PBS. Mice were previously treated with NAD$^+$ for 2 serial days and subsequently infected with *E. coli* by injecting 1 ml of $1\times10^9$ CFU/ml bacterial suspension intraperitoneally. The survival was monitored. In another set of experiments mice were sacrificed 5 hr after infection and kidneys and liver were harvested. The collected tissues were homogenized in 1 ml of sterile PBS and 10-fold serial dilutions plated overnight at 37°C on LB agar plates to determine bacterial load per gram.

## Acknowledgements

JI was supported by the Berlin Institutes of Health Junior Clinician Scientist Program. YN was supported by the Chinese Scholarship Council (201606370196) and Central South University. HRCB was supported by the Swiss Society of Cardiac Surgery. AV was supported by awards from the National Institute of Mental Health (R01MH110438) and National Institute of Neurological Disorders and Stroke (R01NS100808).

## Additional information

### Funding

| Funder | Grant reference number | Author |
|---|---|---|
| Berlin Institutes of Health | Junior Clinician Scientist Program | Jasper Iske |
| China Scholarship Council | 201606370196 | Yeqi Nian |
| Central South University | | Yeqi Nian |
| Swiss Society of Cardiac Surgery | | Hector Rodriguez Cetina Biefer |
| National Institute of Mental Health | R01MH110438 | Anju Vasudevan |
| National Institute of Neurological Disorders and Stroke | R01NS100808 | Anju Vasudevan |

The funders had no role in study design, data collection and interpretation, or the decision to submit the work for publication.

### Author contributions

Jasper Iske, Conceptualization, Data curation, Formal analysis, Validation, Investigation, Visualization, Writing - original draft, Writing – review and editing; Rachid El Fatimy, Yeqi Nian, Data curation, Formal analysis, Validation, Investigation, Visualization, Methodology, Writing - original draft; Amina Ghouzlani, Formal analysis, Investigation, Visualization, Writing – review and editing; Siawosh K Eskandari, Data curation, Formal analysis, Investigation; Hector Rodriguez Cetina Biefer, Investigation,

Writing – review and editing; Anju Vasudevan, Supervision, Investigation, Methodology, Writing – review and editing; Abdallah Elkhal, Conceptualization, Resources, Data curation, Software, Formal analysis, Supervision, Validation, Investigation, Visualization, Methodology, Writing - original draft, Project administration, Writing – review and editing

## Author ORCIDs
Jasper Iske http://orcid.org/0000-0001-8647-3092
Anju Vasudevan http://orcid.org/0000-0001-9814-2389
Abdallah Elkhal http://orcid.org/0000-0002-8320-7722

## Ethics

Blood was obtained from healthy adult volunteers with the only purpose to isolate PBMCs in order to create a basis for macrophage cultures. Blood withdrawal was performed in accordance with guidelines of and approved by the Institutional Review Board of the Charité Universitätsmedizin Berlin (EA4/006/22). Informed consent and consent to publish was obtained from each volunteer in accordance with the Declaration of Helsinki. All personal information collected from volunteers were treated with utmost confidentiality.

This study was performed in strict accordance with the recommendations in the Guide for the Care and Use of Laboratory Animals of the National Institutes of Health. All of the animals were handled according to approved institutional animal care and use committee (IACUC) protocols (#2018N000049) of the Brigham and Women's Hospital.

Joint Public Review: https://doi.org/10.7554/eLife.88686.3.sa1
Author Response https://doi.org/10.7554/eLife.88686.3.sa2

---

# Additional files

## Supplementary files
• MDAR checklist

## Data availability

All data generated or analysed during this study are included in the manuscript as source data files for each figure. Data generated from RNA sequencing of BMDMs have been made publicly available in Dryad (https://doi.org/10.5061/dryad.zw3r228fj).

The following dataset was generated:

| Author(s) | Year | Dataset title | Dataset URL | Database and Identifier |
|---|---|---|---|---|
| Iske J, El Fatimy R, Nian Y, Ghouzlani A, Eskandari SK, Cetina Biefer HR, Vasudevan A, Elkhal A | 2024 | RNA Sequencing of NAD+ and PBS treated Bone Marrow Derived Macrophages following stimulation of the non-canonical Inflammasome | https://doi.org/10.5061/dryad.zw3r228fj | Dryad Digital Repository, 10.5061/dryad.zw3r228fj |

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
