## [Editor Report · eLife assessment]

In this **valuable** contribution, the authors demonstrate that the infusion of NAD+ may prevent death and reduce disease severity from lethal experimental bacterial sepsis, possibly through inflammasome inhibition, without reducing bacterial load. They provide **solid** evidence for these protective effects of NAD+, though the precise mechanisms involved remain unclear and need further support and elucidation. The core findings may well have clinical implications but, in addition to mechanistic clarifications, contextualised interpretation as metabolic adaptation to sepsis would create wider interest.

---

## [Referee Report · Joint Public Review]

Iske et al. provide experimental data that NAD+ lessens disease severity in bacterial sepsis without impacting on the host pathogen load. They show that in macrophages, NAD+ prevents Il1b secretion potentially mediated by Caspase11.

While the in vivo and in vitro data is interesting and hints towards a crucial role of NAD+ to promote metabolic adaptation in sepsis, the manuscript has shortcomings and would profit from several changes and additional experiments that support the claims.

Conceptually, the definition of sepsis is outdated. Sepsis is not SIRS, as in sepsis-2. Sepsis-3 defines sepsis as infection-associated organ dysfunction. This concept needs to be taken into account for the introduction and when describing the potential effects of NAD+ in sepsis. Also, LPS application cannot be considered an appropriate sepsis model, since it only recapitulates the consequence of TLR-4 activation. It is a model of endotoxemia. Also, the LPS data does not allow to draw conclusions about bacterial clearance (L135).

The authors state that protective effects by NAD were independent of the host pathogen load. This clearly indicates that NAD confers protection via enhancing a disease tolerance mechanism, potentially via reducing immunopathology. This aspect is not considered by the authors. The authors should incorporate the concept of disease tolerance in their work, cite the relevant literature on the topic and discuss it their findings in light of the published evidence for metabolic alteration sand adaptations in sepsis.

For the in vitro data, the manuscript would benefit from additional experiments using in vitro infection models.

The figure legend should not repeat the methods and materials section. The nomenclature for mouse protein and genes needs to be thoroughly revised.

L350. The authors write that they dissect the capacity of NAD+ to dampen auto- and alloimmunity. In this work, no data that supports this statement is shown and experiments with autoantigens or alloantigens are not performed. If this refers to another previous publication by the same group, it needs to be put into this context and appropriately cited.

L163 The authors describe pyroptosis but in the figure legend call it apoptosis (Fig.2D). Specific markers for each cell death should be measured and determined which cell death mechanisms is involved.

Animal data comes from an infection model and LPS application. The RNAseq data is obtained from cells primed with Pam3CSK4 and subsequently subjected to LPS. It is unclear how the cell culture model reflects the animal model. As such the link between IFN signaling and the bacterial infection/LPS model are not convincing and need to be further elaborated.

Further experiments with primary cells from Il10 k.o. and Caspase11 k.o. animals should be provided that support the findings in macrophages.

---

## [Author Response]

The following is the authors’ response to the original reviews.

**Reviewer #1 (Public Review):**
The manuscript is very-well written. Although the study is well-conducted the authors should be more convincing on how bacteria residing in tissues do not induce death. The association with IL-10 cytokine production appears weak and more experiments are needed to make it more robust
**Reviewer #2 (Public Review):**
Iske et al. provide experimental data that NAD+ lessens disease severity in bacterial sepsis without impacting on the host pathogen load. They show that in macrophages, NAD+ prevents Il1b secretion potentially mediated by Caspase11.While the in vivo and in vitro data is interesting and hints towards a crucial role of NAD+ to promote metabolic adaptation in sepsis, the manuscript has shortcomings and would profit from several changes and additional experiments that support the claims.Conceptually, the definition of sepsis is outdated. Sepsis is not SIRS, as in sepsis-2. Sepsis-3 defines sepsis as infection-associated organ dysfunction. This concept needs to be taken into account for the introduction and when describing the potential effects of NAD+ in sepsis. Also, LPS application cannot be considered a sepsis model, since it only recapitulates the consequence of TLR-4 activation. It is a model of endotoxemia. Also, the LPS data does not allow to draw conclusions about bacterial clearance (L135).The authors state that protective effects by NAD were independent of the host pathogen load. This clearly indicates that NAD confers protection via enhancing a disease tolerance mechanism, potentially via reducing immunopathology. This aspect is not considered by the authors. The authors should incorporate the concept of disease tolerance in their work, cite the relevant literature on the topic and discuss it their findings in light of the published evidence for metabolic alteration sand adaptations in sepsis.For the in vitro data, the manuscript would benefit from additional experiments using in vitro infection models.In the merge manuscript, the authors provide two different versions of the figures. In one, bar plots are shown without individual data and in the other with scatter blots. All bar plots need to be provided as scatter plots showing individual values.The authors should show further serology data for kidney and liver failure etc. as well as further cytokine data such as IL-6 and TNF to better characterize their models.Careful revision of the entire manuscript, the figure legends and figures is required. The figure legend should not repeat the methods and materials section. The nomenclature for mouse protein and genes needs to be thoroughly revised.L350. The authors write that they dissect the capacity of NAD+ to dampen auto- and alloimmunity. In this work, no data that supports this statement is shown and experiments with autoantigens or alloantigens are not performed.L163 The authors describe pyroptosis but in the figure legend call it apoptosis. Specific markers for each cell death should be measured and determined which cell death mechanisms is involved.Animal data comes from an infection model and LPS application. The RNAseq data is obtained from cells primed with Pam3CSK4 and subsequently subjected to LPS. It is unclear how the cell culture model reflects the animal model. As such the link between IFN signaling and the bacterial infection/LPS model are not convincing and need to be further elaborated.Figure 5: It is unclear how many independent survival experiments were done, how many mice per group were used and whether the difference between groups was statistical significant. This information should be added.Further experiments with primary cells from Il10 k.o. and Caspase11 k.o. animals should be provided that support the findings in macrophages.
**Author Response:**

**Reviewer #1 (Public Review):**
“The manuscript is very-well written. Although the study is well-conducted the authors should be more convincing on how bacteria residing in tissues do not induce death. The association with IL-10 cytokine production appears weak and more experiments are needed to make it more robust.”

Thank you very much for your thoughtful and constructive feedback on our manuscript. We appreciate your positive assessment of the writing quality and the acknowledgment of the wel-lconducted nature of the study.

In regard to the reviewer's comment that "The association with IL-10 cytokine production appears weak," we would like to provide a comprehensive response based on the findings and insights presented in our study (Fig 5). We would like to emphasize several key points to further elucidate this association:

The established knowledge underscores IL-10's capacity to hinder the activation and proliferation of macrophages, thereby safeguarding against an overly aggressive immune-inflammatory reaction (as referenced). In our earlier investigations, we demonstrated that NAD+ orchestrates a systemic generation of IL-10, which assumes a pivotal function in curtailing proinflammatory responses across various conditions, such as autoimmune diseases (as referenced), alloimmunity (as referenced), and bacterial infections (as referenced). In our latest research, we divulge that the introduction of NAD+ leads to an elevated occurrence of IL-10-producing CD4+ T cells, CD8+ T cells, and macrophages, although not dendritic cells (depicted in Figure 5B and C). Furthermore, our comprehensive analyses have substantiated that NAD+ administration thwarts pyroptosis by specifically targeting the non-canonical inflammasome pathway. Intriguingly, our in vitro outcomes suggest that the neutralization of the autocrine IL-10 signaling pathway through a neutralizing antibody and an IL-10 receptor antagonist partially reverses the NAD+-mediated blockage of pyroptosis. These in vitro results imply that NAD+ induces the production of IL-10 cytokines by macrophages, contributing to the suppression of pyroptosis. To corroborate our in vitro conclusions, we employed IL-10 knockout mice and wild-type mice, both treated with either NAD+ or a placebo solution. The wild-type mice treated with NAD+ displayed a survival rate exceeding 80%, whereas the IL-10 knockout mice exhibited a survival rate of "only" 40%. These in vivo findings align with our in vitro discoveries, underscoring the crucial role of NAD+mediated IL-10 cytokine production in impeding pyroptosis through NAD+ and shielding against septic shock. Drawing from our prior and current investigations, we respectfully disagree with the reviewer's characterization of our work as "weak."

Recommendations for the authors‘’I suggest that animals subject to *E. coli* infection need to be followed-up for longer and sacrificed at a later time points. It is too difficult to believe that mice are surviving with full resting bacteria in tissues. Do results suggest a full shut-down of the mechanism? What was the level of infiltration of the tissues by neutrophils?’’‘’I have difficulty to agree with the survival results of the IL-10(-/-) mice of Figure 5E. Can the authors provide the p-values and follow-up for longer? Why the WT and the IL-10(-/-) mice survive the same?’’

Thank you for your thoughtful and constructive comments on our manuscript. We appreciate your valuable insights, and we have carefully considered your suggestions.

We thank the reviewers for this comment. We have indeed followed-up for a longer period of time mice subjected to *E. coli* infection and LPS (54mg/kg). Mice infected and treated with NAD+ survived for several months and recovered fully after 10 days. Mice survived for at least a year following infection. We have now included a sentence regarding the long-term survival in the results section of Figure 1 entitled “NAD+ protects mice against septic shock not via bacterial clearance but via inflammasome blockade”. Figure illustrating the level of infiltration of the tissues by neutrophils was added in supplementary data as supplementary figure 4.

In contrast, WT and IL-10-/- mice failed to withstand *E. coli* or LPS (54mg/kg) administration when treated with a placebo solution. To our knowledge, our investigation represents the pioneering instance of successfully conferring protection against the lethal doses of *E. coli* and LPS administered to animals. Considering the potent immunosuppressive nature of IL-10, our anticipation was that IL-10-/- mice would manifest an exacerbated inflammatory response subsequent to LPS administration, in contrast to WT mice. Our in vivo findings indeed corroborate this assumption, revealing that IL-10-/- mice succumbed more swiftly to LPS administration, displaying statistically significant disparities in survival rates compared to WT mice (p value of 0.0154). The pertinent p-value has been thoughtfully included in Figure 5E of our study.

**Reviewer #2 (Public Review):**
“Iske et al. provide experimental data that NAD+ lessens disease severity in bacterial sepsis without impacting on the host pathogen load. They show that in macrophages, NAD+ prevents Il1b secretion potentially mediated by Caspase11.While the in vivo and in vitro data is interesting and hints towards a crucial role of NAD+ to promote metabolic adaptation in sepsis, the manuscript has shortcomings and would profit from several changes and additional experiments that support the claims.Conceptually, the definition of sepsis is outdated. Sepsis is not SIRS, as in sepsis-2. Sepsis-3 defines sepsis as infection-associated organ dysfunction. This concept needs to be taken into account for the introduction and when describing the potential effects of NAD+ in sepsis. Also, LPS application cannot be considered a sepsis model, since it only recapitulates the consequence of TLR-4 activation. It is a model of endotoxemia. Also, the LPS data does not allow to draw conclusions about bacterial clearance (L135).The authors state that protective effects by NAD were independent of the host pathogen load. This clearly indicates that NAD confers protection via enhancing a disease tolerance mechanism, potentially via reducing immunopathology. This aspect is not considered by the authors. The authors should incorporate the concept of disease tolerance in their work, cite the relevant literature on the topic and discuss it their findings in light of the published evidence for metabolic alteration sand adaptations in sepsis.For the in vitro data, the manuscript would benefit from additional experiments using in vitro infection models.In the merge manuscript, the authors provide two different versions of the figures. In one, bar plots are shown without individual data and in the other with scatter blots. All bar plots need to be provided as scatter plots showing individual values.The authors should show further serology data for kidney and liver failure etc. as well as further cytokine data such as IL-6 and TNF to better characterize their models.Careful revision of the entire manuscript, the figure legends and figures is required. The figure legend should not repeat the methods and materials section. The nomenclature for mouse protein and genes needs to be thoroughly revised.L350. The authors write that they dissect the capacity of NAD+ to dampen auto- and alloimmunity. In this work, no data that supports this statement is shown and experiments with autoantigens or alloantigens are not performed.L163 The authors describe pyroptosis but in the figure legend call it apoptosis. Specific markers for each cell death should be measured and determined which cell death mechanisms is involved.Animal data comes from an infection model and LPS application. The RNAseq data is obtained from cells primed with Pam3CSK4 and subsequently subjected to LPS. It is unclear how the cell culture model reflects the animal model. As such the link between IFN signaling and the bacterial infection/LPS model are not convincing and need to be further elaborated.Figure 5: It is unclear how many independent survival experiments were done, how many mice per group were used and whether the difference between groups was statistical significant. This information should be added.Further experiments with primary cells from Il10 k.o. and Caspase11 k.o. animals should be provided that support the findings in macrophages.”

Thank you for taking the time to review our manuscript. We appreciate your insightful comments and valuable feedback regarding our study on the role protective role and underlying mechanisms of NAD+ in septic shock.

“While the in vivo and in vitro data is interesting and hints towards a crucial role of NAD+ to promote metabolic adaptation in sepsis, the manuscript has shortcomings and would profit from several changes and additional experiments that support the claims.”

We would like to point out that our current study does not underscore a metabolic adaptation in sepsis but more an immune regulation and a specific blockade of the non-canonical inflammasome signaling machinery.

“Conceptually, the definition of sepsis is outdated. Sepsis is not SIRS, as in sepsis-2. Sepsis-3 defines sepsis as infection-associated organ dysfunction. This concept needs to be taken into account for the introduction and when describing the potential effects of NAD+ in sepsis. Also, LPS application cannot be considered a sepsis model, since it only recapitulates the consequence of TLR-4 activation. It is a model of endotoxemia. Also, the LPS data does not allow to draw conclusions about bacterial clearance (L135).”

Our study uses highly lethal doses of *E. coli* or LPS. These doses have been shown to result in multiple organ failure (1, 2). For many decades until now an un-numerable number of studies have used LPS as a model of sepsis (3, 4, 5). We have used LPS animal model based on a study published in 2013 by Kayagaki et al. (1), where the authors reported a novel TLR4-independent mechanism but mediated via activate caspase-11. We used the same animal model to demonstrate the specific role of NAD+ in targeting this TLR4-independent mechanism but mediated via activate caspase-11 and underscore NAD+’s mode of protection.

Moreover, we have not only used LPS but bacterial infection as well using *E. coli*. We have also previously published an additional research article demonstrating the protective effect against Listeria Monocytogenes (6). The only model we currently did not use in our current study, is a cecal ligation puncture (CLP) model which is also another common animal model for sepsis.

Our conclusions regarding bacterial clearance are based not only on LPS results but also based on the bacterial load measurement and survival (Figure 1B&C) following *E. coli* administration in different tissues (kidney and liver) and not LPS.

“The authors state that protective effects by NAD were independent of the host pathogen load. This clearly indicates that NAD confers protection via enhancing a disease tolerance mechanism, potentially via reducing immunopathology. This aspect is not considered by the authors. The authors should incorporate the concept of disease tolerance in their work, cite the relevant literature on the topic and discuss it their findings in light of the published evidence for metabolic alteration sand adaptations in sepsis.”

We respectfully disagree with the reviewer’s comment and do not believe that NAD+ enhances disease tolerance. We have supporting data indicating that NAD+ mediates protection via a specific blockade of the non-canonical inflammasome pathway, which prevents an over-zealous immune response that results in organ damage and multiple organ failure (MOF). Moreover, we demonstrate that not only NAD+ mediates protection via a specific blockade of the non-canonical inflammasome pathway but prevents septic shock induced death by an additional immunosuppression mediated by the systemic production of IL-10.

Both Caspase-11 and IL-10 pathways are crucial in NAD+ mediated protection against lethal doses of *E. coli* and LPS administration. Figure 5A indicates that caspase-11-/- mice treated with PBS have a modest survival rate (~40% survival) when compared to the group of mice treated with NAD+ (>80% survival). These data indicate that NAD+ promotes survival via a caspase-11independent mechanism. Similarly, wild type mice subjected to NAD+ administration exhibited >80% survival, while NAD+ administration to IL-10-/- mice resulted only in a 40% survival rate. Based on these findings, we believe that NAD+ mediated protection against septic shock via a blockade of caspase-11 blockade and by IL-10 cytokine production that dampened the overzealous immune response rather than a disease tolerance.

“For the in vitro data, the manuscript would benefit from additional experiments using in vitro infection models.”

In the current study we have used two in vivo models using LPS and *E. coli* a gram-negative bacterium. We have also previously reported the protective role of NAD+ in the context of Listeria Monocytogenes (6) a gram-positive bacterium. In the current study, our aim was to demonstrate the inhibitory role of NAD+ on the non-canonical pathway specifically. We believe that additional in vitro experiments for this study are out of scope.

“In the merge manuscript, the authors provide two different versions of the figures. In one, bar plots are shown without individual data and in the other with scatter blots. All bar plots need to be provided as scatter plots showing individual values.”

As requested by reviewer #2 all bar plots are now provided as scatter plots showing individual values.

“The authors should show further serology data for kidney and liver failure etc. as well as further cytokine data such as IL-6 and TNF to better characterize their models.”

We did not perform further serology analysis, but we did measure IL-6 and TNFα in mice treated with NAD+ or PBS. Mice treated with NAD+ had a reduced systemic level of both cytokines IL-6 and TNFα. We have now added the figures (Figure 1F). In addition, we performed a long-term survival, and all mice treated with NAD+ recovered fully after 10 days and survived over a year after infection. In addition, the mice that survived following NAD+ treatment died of old age.

“Careful revision of the entire manuscript, the figure legends and figures is required. The figure legend should not repeat the methods and materials section. The nomenclature for mouse protein and genes needs to be thoroughly revised.”

A Careful revision of the entire manuscript has been performed.

“L350. The authors write that they dissect the capacity of NAD+ to dampen auto- and alloimmunity. In this work, no data that supports this statement is shown and experiments with autoantigens or alloantigens are not performed.”

We thank the reviewer for this comment. We have now re-phrased our last sentence in the discussion and included references for our previous work. We have now stated:” We have previously reported that NAD+ administration can block auto- (7) and allo-immunity (8) via IL10 cytokine production. Here, we unveiled the capacity of NAD+ to protect against sepsisinduced death via a specific blockade of the non-canonical inflammasome pathway and a robust immunosuppression mediated by IL-10 cytokine production.

L163 The authors describe pyroptosis but in the figure legend call it apoptosis. Specific markers for each cell death should be measured and determined which cell death mechanisms is involved.

We thank the reviewer for this comment. We have focuses on pyoptosis-mediated cell death and not apoptosis. We have now replaced the term “apoptosis” by “pyroptosis-mediated to cell death”.

“Animal data comes from an infection model and LPS application. The RNAseq data is obtained from cells primed with Pam3CSK4 and subsequently subjected to LPS. It is unclear how the cell culture model reflects the animal model. As such the link between IFN signaling and the bacterial infection/LPS model are not convincing and need to be further elaborated.”

Our findings, depicted in Figure 3, pertain exclusively to in vitro investigations rather than in vivo examinations. Our research has demonstrated the selective inhibition of the non-canonical inflammasome pathway by NAD+, with a primary focus on unraveling the specific signaling pathway influenced by NAD+. Our in vitro outcomes indicate that the introduction of recombinant IFN-β counteracted the inhibitory effect of NAD+ on the non-canonical pathway. However, it's important to note that we have not evaluated the IFN-β pathway within our *E. coli* and LPS in vivo models. Our primary intention was to exclusively decipher the roles of IFN-β and NAD+ in the context of inhibiting the non-canonical inflammasome, without extending our investigation to the broader in vivo scenarios.

“Figure 5: It is unclear how many independent survival experiments were done, how many mice per group were used and whether the difference between groups was statistical significant. This information should be added.”

We have now included the number of experiments, p values and number of animals used in Figure 5.

“Further experiments with primary cells from Il10 k.o. and Caspase11 k.o. animals should be provided that support the findings in macrophages.”

We concur with the reviewer's suggestion regarding the need for further experiments involving primary cells from IL-10-/- and Caspase-11-/- mice. However, we are uncertain about the potential contribution of these experiments in generating novel or supplementary findings to the existing study.

Recommendations For The Authors:Besides the comments made in the public section, there are further issues that need to be considered by the authors.“It is unclear what signifies „impressive, L106" or „dramatic, L257"”

“impressive” meant that we were surprised by the results since to the best of our knowledge prior this study there exists no report/study claiming such survival (>80%) following such high dose of *E. coli*. In this aspect protective effects of NAD+ are unique.“dramatic” We (8) and others (9, 10) have previously used this term to describe a robust increase of cytokine production.

“L116. The authors describe „symptoms". It should be clarified what symptoms they observed and the data should be shown. If only temperature is available, then this should be said. It would be interesting to see effects of NAD+ on the glucose levels of the animals during sepsis.”

We thank the reviewer’s comment. We have measured only temperature. We believe that glucose level is beyond the scope of this study.

“L29. Sepsis is not restricted to bacterial and viral pathogens. Also fungi and protozoa can cause sepsis.”

We have now included fungi and protozoa.

“Suppl.Fig.1. A scale should be added.”

Scale has been added

“L822. Lethal dose of LPS would mean that this was lethal for all mice. However, the data suggests that NAD+ treated animals would not have died. This should be clarified.”

Here we meant lethal dose in absence of NAD+ treatment. Our study focuses on the protective role of NAD+ in a lethal context (bacterial and LPS).

“L823/824. The part of the sentence: ... IHC was performed staining for H&E.. is incomplete.”

We thank the reviewer’s comment. We have re-phrased our sentence.

“L804. IL-10 is not a pathway. This should be revised.”

We have replaced “pathway” by” mechanism”.

“The graphical abstract should be the last figure summarizing all findings.”

Figure 4 isn't the final illustration, as it doesn't encompass an overarching graphical summary of our discoveries. Instead, it exclusively highlights the findings related to NAD+'s impact on noncanonical inflammasome inhibition. Notably, this figure omits NAD+-mediated IL-10 cytokine generation and its crucial role in mitigating septic shock.

“The authors report that they used a dosage of 54mg/kg LPS (l.502). This is a rather unusual concentration. How was this determined?”

This was initially based on the first study reporting the role of casapase-11 in septic shock induced death published in 2013 by Kayagaki et al. (1). Many other have used this dosage for septic shock induced death animal model (11, 12, 13).

References:

1. Kayagaki N, et al. Noncanonical inflammasome activation by intracellular LPS independ ent of TLR4. Science 341, 1246‐1249 (2013).

2. Qin, X., Jiang, X., Jiang, X. et al. Micheliolide inhibits LPS-induced inflammatory response and protects mice from LPS challenge. Sci Rep 6, 23240 (2016).

3. Li Z, Qu W, Zhang D, Sun Y, Shang D. The antimicrobial peptide chensinin-1b alleviates the inflammatory response by targeting the TLR4/NF-κB signaling pathway and inhibits *Pseudomonas aeruginosa* infection and LPS-mediated sepsis. Biomed Pharmacother.2023 Aug 1; 165:115227.

4. Ramani V, Madhusoodhanan R, Kosanke S, Awasthi S. A TLR4-interacting SPA4 peptide inhibits LPS-induced lung inflammation. Innate Immun. 2013 Dec;19(6):596610.

5. Zhang Y, Lu Y, Ma L, Cao X, Xiao J, Chen J, Jiao S, Gao Y, Liu C, Duan Z, Li D, He Y, Wei B, Wang H. Activation of vascular endothelial growth factor receptor-3 in macrophages restrains TLR4-NF-κB signaling and protects against endotoxin shock. Immunity. 2014 Apr 17;40(4):501-14.

6. Rodriguez Cetina Biefer H, Heinbokel T, Uehara H, Camacho V, Minami K, Nian Y,Koduru S, El Fatimy R, Ghiran I, Trachtenberg AJ, de la Fuente MA, Azuma H, Akbari O, Tullius SG, Vasudevan A, Elkhal A. Mast cells regulate CD4+ T-cell differentiation in the absence of antigen presentation. J Allergy Clin Immunol. 2018 Dec;142(6):18941908.e7.

7. Tullius SG, Biefer HR, Li S, Trachtenberg AJ, Edtinger K, Quante M, Krenzien F, Uehara H, Yang X, Kissick HT, Kuo WP, Ghiran I, de la Fuente MA, Arredouani MS,Camacho V, Tigges JC, Toxavidis V, El Fatimy R, Smith BD, Vasudevan A, ElKhal A.NAD+ protects against EAE by regulating CD4+ T-cell differentiation. Nat Commun.2014 Oct 7;5:5101.

8. Elkhal A, et al. NAD(+) regulates Treg cell fate and promotes allograft survival via a systemic IL‐10 production that is CD4(+) CD25(+) Foxp3(+) T cells independent. Sci Rep 6, 22325 (2016).

9. Natalia Garcia-Becerra, Marco Ulises Aguila-Estrada, Luis Arturo Palafox-Mariscal,Georgina Hernandez-Flores, Adriana Aguilar-Lemarroy, Luis Felipe Jave-Suarez,FOXP3 Isoforms Expression in Cervical Cancer: Evidence about the Cancer-RelatedProperties of FOXP3Δ2Δ7 in Keratinocytes, Cancers, 15, 2, (347), (2023).

10. Estelle Bettelli, Maryam Dastrange, Mohamed Oukka. Foxp3 interacts with nuclear factor of activated T cells and NF-κB to repress cytokine gene expression and effector functions of T helper cells. Proceedings of the National Academy of Sciences. 2005.102; 14; 5138-5143.

11. Han Gyung Kim, Chaeyoung Lee, Ji Hye Yoon, Ji Hye Kim, Jae Youl Cho,BN82002 alleviated tissue damage of septic mice by reducing inflammatory response through inhibiting AKT2/NF-κB signaling pathway,Biomedicine & Pharmacotherapy,Volume 148,2022,112740.

12. Tao Q, Zhang Z-D, Qin Z, Liu X-W, Li S-H, Bai L-X, Ge W-B, Li J-Y and Yang Y-J (2022) Aspirin eugenol ester alleviates lipopolysaccharide-induced acute lung injury in rats while stabilizing serum metabolites levels. Front. Immunol. 13:939106.

13. Chen, N, Ou, Z, Zhang, W, Zhu, X, Li, P, Gong, J. Cathepsin B regulate non-canonical NLRP3 inflammasome pathway by modulating activation of caspase-11 in Kupffer cells. Cell Prolif. 2018; 51:e12487.